# Direct Ink Writing Glass: A Preliminary Step for Optical Application

**DOI:** 10.3390/ma13071636

**Published:** 2020-04-01

**Authors:** Bo Nan, Przemysław Gołębiewski, Ryszard Buczyński, Francisco J. Galindo-Rosales, José M. F. Ferreira

**Affiliations:** 1Department of Materials and Ceramic Engineering, University of Aveiro, CICECO—Aveiro Materials Institute, 3810-193 Aveiro, Portugal; Bo.Nan@ceitec.vutbr.cz; 2CEITEC-Central European Institute of Technology, Brno University of Technology, Purkynova 656/123, 612 00 Brno, Czech Republic; 3Institute of Electronic Materials Technology, Wólczyńska 133, 01-919 Warsaw, Poland; przemyslaw.golebiewski@itme.edu.pl (P.G.); ryszard.buczynski@fuw.edu.pl (R.B.); 4Faculty of Physics, University of Warsaw, Pasteura 5, 02-093 Warsaw, Poland; 5CEFT, Department of Chemical Engineering, Faculty of Engineering of the University of Porto, 4200-465 Porto, Portugal; galindo@fe.up.pt

**Keywords:** direct ink writing, glass, rheology

## Abstract

In this paper, we present a preliminary study and conceptual idea concerning 3D printing water-sensitive glass, using a borosilicate glass with high alkali and alkaline oxide contents as an example in direct ink writing. The investigated material was prepared in the form of a glass frit, which was further ground in order to obtain a fine powder of desired particle size distribution. In a following step, inks were prepared by mixing the fine glass powder with Pluoronic F-127 hydrogel. The acquired pastes were rheologically characterized and printed using a Robocasting device. Differential scanning calorimetry (DSC) experiments were performed for base materials and the obtained green bodies. After sintering, scanning electron microscope (SEM) and X-ray diffraction (XRD) analyses were carried out in order to examine microstructure and the eventual presence of crystalline phase inclusions. The results confirmed that the as obtained inks exhibit stable rheological properties despite the propensity of glass to undergo hydrolysis and could be adjusted to desirable values for 3D printing. No additional phase was observed, supporting the suitability of the designed technology for the production of water sensitive glass inks. SEM micrographs of the sintered samples revealed the presence of closed porosity, which may be the main reason of light scattering.

## 1. Introduction

After almost thirty years of development, additive manufacturing, or commonly named 3D printing, has started to play an important role in the modern industry such as automobile, aerospace, bioengineering, and so on, and the interest of research still maintains enthusiasm [1,2,3,4,5]. It is not only a near-net shaping, but also a moldless shaping method, that can be readily categorized into two different types, namely light-based and ink-based 3D printing [6], or into seven different types, according to the classification of American Society for Testing and Materials (ASTM) 52900:2015 standard [1]. Among those advanced techniques, most of the studies have been focused on metals, ceramics or polymers [7,8,9,10,11], while only a few attempts are applied in shaping glass.

Similar to many materials that are hard to machine, the difficulty of shaping glass lies in the material itself, as it is brittle and usually sensitive to thermal shock. Many types of glasses are also prone to crystallization upon reheating, which limits the processing temperature range. Moreover, it can be difficult to apply shaping techniques for glasses that are based on powder densification, as the main issue is its behavior in elevated temperatures. Unlike sintering ceramics or metals, a glass starts to flow as a viscous material after reaching the transition temperature. As a result, it is challenging to retain the shape of an already formed object while the sintering process is applied, otherwise a very precise control of temperature and dwelling time is needed. Apart from the solid-state processing, colloidal shaping processes require well-dispersed suspensions. However, the complex compositions of various types of glasses may undergo preferential ionic leaching of certain components in the dispersing liquid that affects the homogenization of the dispersed system. Such compositions commonly consist of not only glass-forming compounds, but also additives to either modify final parameters of glass or reduce manufacturing cost. Taking the water-based system as an example, alkaline earth and alkali oxides such as Na_2_O, which are aimed to lower the operating temperature range for silica-based glasses, may be hydrolyzed, leading to the alteration of rheological properties of the suspension and further inhibiting the liquid processing, as a result of pH change in the glass suspensions with time. Although using non-aqueous suspension can eliminate the dissolving or hydrolyzing problems, the organic liquids are usually toxic and more expensive. Therefore, research on liquid processing glass, especially connected with additive manufacturing, focuses mainly on silica-rich [12,13] and bioactive glasses [14,15], as the first type is more resistant to thermal shock and allows the production of dense and transparent samples, and the latter type is preferred in porous scaffold formed to enhance its bioactivity and osteointegration potential.

As 3D printing is a readily accessible shaping method and has been utilized in many kinds of materials, it may be still a good candidate for shaping glass as well. Recent reports have brought excellent examples and gained confidence in both the glass and 3D printing community by various methods such as depositing and annealing the molten filaments, laser aided filament-fed process, direct ink writing, stereolithography, and digital light processing [12,13,16,17,18,19]. Among them, the first two require a high energy source such as heating or laser assisted printing, with the glass being prepared in one step, while at least two steps are required for the other methods, that is to say, green glass samples were printed with an extra follow-up sintering process. Although it seems that the one-step procedure is an easy option, it involves very precise control of temperature and expensive assisting parts to guarantee the stability of the printing process. For printing in more steps, it is easy to obtain more broad compositions and increase the resolution of the as-designed geometry by utilizing and changing the formulation of the liquid molecular precursor, being economically applicable for a small batch of samples at a laboratory level. The comparison of these reported methods can be found in Table 1.

Here, we report a preliminary experiment on printing borosilicate glass by direct ink writing. Unlike the previous reports in [12,16], the micron-sized borosilicate glass particles are utilized instead of the nanoparticles or liquid molecular precursors (sol–gel) of silica or silica-based compositions. The first challenge is to form colloidal stable inks as the borosilicate glass used in this study undergoes hydrolysis, which, as mentioned before, makes suspension rheologically unpredictable. In this study, we proposed an ink prepared by dispersing glass powder in the Pluronic F-127 hydrogel. Pluronic F-127 is a triblock copolymer of polyethylene oxide-polypropylene oxide-polyethylene oxide (PEO-PPO-PEO). Because of the hydrophilic nature of PEO and hydrophobic nature of PPO, Pluronic exhibits amphiphilic behavior. As a result of hydrophobic effect described in [20], after reaching a certain concentration named critical micelle concentration (CMC), it starts to form micelles, typically in a spherical shape. In addition, this hydrophobic effect corresponds to an increase of entropy. It implies that an increase in temperature will result in a higher degree of micellization. The gel is formed because of the increasing volume of the micelles and repulsive forces between them [21]. In previous studies, Bromberg et al. [22] and Park et al. [23] reported that Pluronic hydrogels are kept stable even when the pH of the solution is changed. On the basis of this data, it is assumed that the ionic strength insensitivity of the Pluronic F-127 hydrogel-based suspensions confers good stability to their rheological properties, thus facilitating the printing of water-sensitive glass materials. The second challenge is related to the thermal treatment processes: debinding (burnout of organic substances present in the printed sample) and sintering, which should result in obtaining denser and more transparent printed samples. If both challenges are overcome, it will be possible to print more complex shapes, for example, preforms for optical fiber production.

## 2. Materials and Methods

### 2.1. Preparation of Glass Powder

The borosilicate glass considered in this study was prepared in the form of frit. The following fine powders were used as raw materials: SiO_2_ (milled fused silica, Lianyungang DIGHEN Composite Material Technology Co.,Ltd, Lianyungang, China), H_3_BO_3_ (Chempur, Piekary Śląskie, Poland, ≥99.5%), MgO (Chempur, Poland, ≥98%), CaCO_3_ (Chempur, Poland, ≥99%), Na_2_CO_3_ (Chempur, Poland, ≥98.8%), KNO_3_ (Chempur, Poland, ≥99%), K_2_CO_3_ (Chempur, Poland, ≥99%). The composition of the fabricated glass is presented in Table 2.

Calculations were made in order to obtain 750 g of glass frit. In the first step, proper amounts of raw materials were mixed using alumina mortar. The obtained mixture was transferred to platinum crucible, which was preheated to 1250 °C in the furnace. Afterwards, the batch was melted at 1370 °C for 3 h. During the melting process, the batch was mixed three times using silica rod. In the next step, melt was cast into cold water. As-cast frit was separated from water and dried in the oven at 60 °C for 24 h. Subsequently, received material was mechanically crushed into smaller pieces using alumina mortar, as well as being milled in a planetary mill (PM) using agate container and balls (mixture of different sizes, ball-to-powder ratio of 2:1). Small portions of powder after 2, 4, and 6 h of planetary milling were collected. Finally, glass was attrition milled (AM) in ethanol (zirconia balls with 3 mm diameter and the ball-to-powder ratio of 3:1) at the rate of 500 rpm for 3 h, in order to further break the large particles into smaller ones. After three hours, a small portion of suspension was collected and dried at 80 °C overnight. Particle size and particle size distributions of gathered borosilicate glass powders were determined using a laser diffraction analysis (Coulter LS 230, Brea, CA, USA). A given amount of powder (~0.3 g) was dispersed in 50 mL of deionized water with the aid of a drop of Dolapix CE64 (ZSCHIMMER & SCHWARZ, Lahnstein, Germany). The dilute suspension was sonicated for 5 min to destroy the agglomerates before performing the particle size and particle distribution measurements. Finally, suspension after attrition milling was dried at 80 °C overnight and passed through a 40 µm polymer sieve.

### 2.2. Ink Preparation

The glass inks for direct ink writing were prepared by following several successive steps, as depicted in Figure 1. Before preparing glass inks, Pluronic F-127 (Sigma-Aldrich, Darmstadt, Germany) was dissolved into deionized water with concentrations of 10 wt.%, 20 wt.%, 30 wt.%, 40 wt.%, and 50 wt.%, respectively. The mass of the chemicals was weighed in a high-accuracy (0.0001 g) weighing scale. All the polymer solutions were kept in hermetic containers to prevent water evaporation, and stored in a fridge (~4 °C) overnight, in order to further homogenize the composition.

Glass inks with a series of solids loading of 15.0 wt.%, 25.0 wt.%, and 35.0 wt.% were prepared by mixing the as-milled glass powder with the Pluronic stock solution (30 wt.%) in a planetary mixer (THINKY ARE-250, Tokyo, Japan). The powder was gradually added into the polymer solution, followed by stirring at the rate of 1500 rpm until the entire objective amount of the powder was incorporated. During each mixing interval, the as-mixed suspension was immediately taken out from the mixer and kept into an iced water tank to cool it down for 3 min. In contrast to the preparation procedures previously reported, involving the successive mixing steps after adding each kind of processing additive (dispersant-binder-coagulant) [24,25,26], the one-step mixing utilized in this study simplifies and facilitates the preparation of the inks. The final as-mixed suspension was sonicated for 5 min to destroy the possible agglomerates prior to any further use.

### 2.3. Rheological Tests

The viscoelastic properties of the Pluronic F-127 solutions and glass inks were characterized by a rotational rheometer (Anton Paar MCR301, Graz, Austria) equipped with a controlled shear stress (CSS) rotational module and a direct strain oscillatory module (DSO). Steady-state flow curves were obtained by imposing a logarithmic ramp in shear stress. The limits of the linear viscoelastic region (LVE region) were determined by means of amplitude strain sweep experiments at the constant frequency of 1.0 Hz. Additionally, three interval thixotropy tests (3ITT) were developed in order to determine if the samples are prone to quick recovery of the gel state after being sheared at large deformations, which is paramount for ensuring the shape retention of the extruded filaments [27]. The 3ITT experiments consisted of three concatenated oscillatory shear deformation steps at a constant frequency of 1.0 Hz: the first interval consisted of an of 0.005% during 120 s; the second step at 500% for 120 s; and the third interval at the same deformation as the first one during 1560 s. On the basis of the result obtained from amplitude sweep experiments, oscillations with amplitude of deformation of 0.005% were found within the LVE region, while a shear strain of 500% was found in the non-linear viscoelastic region.

To minimize the possible slipping effect, measurements were performed with a serrated plate-and-plate (PP10/P2, Anton Paar, Graz, Austria) with a gap of 0.4 mm. The temperature was set at 25 °C and controlled with a built-in Peltier system, and a solvent trap cover was used to prevent a partial evaporation of the solvent from the samples during the tests.

### 2.4. Direct Ink Writing

Three kinds of as-prepared ink (with solids loading of 15 wt.%, 25 wt.% and 35 wt.%) were loaded into syringes (Luer locker, 3 mL, Nordson, Westlake, OH, USA) and extruded through the same nozzle (Optimum^®^ general purpose dispense tips, internal diameter 250 µm, Nordson, USA). The hollow cylinder geometry was designed by a CAD software (FreeCAD, version 0.18) and sliced by Robocad software attached to the Robocasting system (Model EBRD-A32, 3D Inks, LLC, Tulsa, OK USA). The details of equipment and its related accessories utilized during experiments are shown in Figure 2. Before printing, the printing atmosphere (temperature ~25 °C and relative humidity ~80%) was adjusted by an air humidifier (SHF 911GR, Sencor, Ricany Czech Republic) and monitored by a thermo-hygrometer (NK-TH2, NKTECH, Shenzhen, China). The syringe driver dispensed the ink on an aluminum oxide substrate (surface spread with a thin layer of anti-stick agent, 100 × 100 × 1 mm, Zhengzhou Kejia Furnace CO., LTD, Zhengzhou, China). After printing each sample, the substrate was transferred immediately into a homemade box with wet tissue inside and dried in a preheated oven at 80 °C until the samples dried and hardened. The samples could be easily detached from the substrate after drying, and the greasy anti-stick layer and Pluronic could be removed by further firing process, with the ramp rate 1 °C/min up to 500 °C, holding at this temperature for 2 h.

### 2.5. Differential Scanning Calorimetry Coupled with Thermogravimetry (DSC/DSC-TG)

To evaluate suitable parameters for the debinding and sintering processes of the printed glass samples, differential scanning calorimetry coupled with thermogravimetry was performed. Green body received after printing and drying was crushed and ground in agate mortar to obtain powder. Measurement was carried out in alumina crucible in the temperature range from 25 °C to 1100 °C at a heating rate of 10 °C/min using Netzsch STA 449F1 calorimeter. Second alumina crucible was used as a reference. Measurements of glass frit and pure Pluronic F-127 were also made to distinguish processes corresponding to each material. In case of Pluronic F-127, the measurement was carried out with slower heating rate equal to 2 °C/min up to 1000 °C. All of the above measurements were performed in “synthetic air” atmosphere with the composition of 80 vol.% of Ar and 20% of O_2_.

### 2.6. Sintering

After drying, the printed samples were fired in a two-step process according to the results obtained from DSC-TG measurement. Firstly, samples were debinded at 500 °C for 8 h (heating rate 30 °C/h) and afterwards sintered at 720 °C (heating rate 150 °C/h) for 30 min.

### 2.7. X-ray Powder Diffraction (XRD)

In order to determine the presence of crystalline impurities introduced during samples elaboration, especially during milling process (quartz or/and zirconia), XRD measurement of the sintered glass sample (crushed and ground to obtain fine powder) was performed. In this study, Bruker D8 Advance X-ray diffractometer was used. Measurement was carried out in symmetric geometry, with Cu anode (λ_CuKα_ = 1.5418 Å) using Lynx Eye type high efficiency linear detector. Samples were tested in 2θ~10–100° at a step rate of 0.02° and dwelling time equal to 2 s.

### 2.8. Scanning Electron Microscopy (SEM)

Printed and sintered rings were broken on purpose in order to observe their fractures. Next, specimens were covered with a thin (~15 nm) carbon layer using thin film deposition system (PVD 75, Kurt J. Lesker Co., Jefferson Hills, PA, USA). Investigation of obtained samples’ microstructure was carried out using Sigma 500 (Carl Zeiss, Oberkochen, Germany) scanning electron microscope.

## 3. Results

Figure 3 and Table 3 present particle size distributions of glass powders obtained after different milling times. Samples collected after 2 h of planetary milling cannot reach the necessary signal threshold, which means that the milling time was insufficient and the size of most particles was still too high. One can observe that increasing the time of milling gradually results in smaller particle sizes with narrower distributions. After 4 h of planetary milling, a broad plateau from 4 to 40 µm can be identified. Subsequent milling for additional 2 h results in a bimodal particle size distribution with two maxima at around 3 and 20 µm. Finally, after applying attrition milling, the powder starts to show almost unimodal distribution with a median particle size, d_50_ = 4.2 µm (meaning that half of the particle’s population lies below this value). Similarly, other d_x_ numbers mean that *x* percent of the particle’s population lies below this specific value. Moreover, d_10_ and d_25_ values remain at a similar level for the whole milling time, while d_90_ values diminish over three times. This means that the milling process under such a condition does not significantly affect particles with a size below 2 µm.

Preliminary experiments were firstly carried out using the common approach for preparing inks comprising three processing additives, that is, anionic dispersant, thickener (hydroxypropyl methylcellulose, HPMC), and cationic coagulant (polyethyleneimine, PEI). However, it soon became clear that this approach could not be successfully applied to this glass system. The electrostatic interactions between an anionic dispersant and the cationic coagulant (PEI) are at a maximum within the neutral pH region. As the tendency of the glass powder to hydrolyze is high, the alkaline earth and alkali ions leached to the dispersing water cause the pH of the suspension to increase to high values at which the PEI species exist essentially in the non-dissociated form, thus preventing the coagulation phenomenon from occurring [15]. Under such conditions, the stiffness of the ink is insufficient to grant shape retention to the extruded filaments, and the as-printed layers are not mechanically strong to support the consecutively deposited layers while building up the part. Instead, the gelation of Pluronic F-127 is driven by the temperature-dependent hydrophilic−hydrophobic interactions [20,21], which are less sensitive to the ionic strength of the dispersing liquid. This is to say that the rheological behavior of glass powder inks prepared with Pluronic hydrogel is not much affected by glass hydrolysis.

Another advantage of using Pluronic hydrogel, which does not involve the addition of processing additive solutions inevitably causing a diluting effect, is the ease with which the required solids volume fraction can be accurately adjusted and controlled. Pluronic solutions with five different concentrations were prepared. They can readily liquify at a low temperature and jellify again when placed at room temperature [28]. The solution with 10 wt.% Pluronic is able to flow even at room temperature, thus being improper for further glass ink gelation and printing, which is why it was not tested on rheometer. Figure 4 depicts the results of flow curves measured for both pure Pluronic solutions and glass inks. With an increasing the concentration, the viscosity of the Pluronic solutions and of the glass inks increases from 2 × 10^3^ Pa·s to 3 × 10^4^ Pa·s at a low shear rate of 10^−1^ s^−1^; and from 2 × 10^4^ Pa·s to 3 × 10^5^ Pa·s at 10^−1^ s^−1^, respectively, as shown in Figure 4a,c. The shear rate–shear stress curves (Figure 4b,d) reveal that the curves become flatter with increasing concentrations of both Pluronic and glass in the inks, as a mechanically percolating and space filling network is developed, which is characteristic of a solid-like material. The yield stress (σ_y_) can be calculated by extrapolating the steady-state flow curve to zero shear rate [29]. The higher concentration of the solutions/suspensions will increase the yield stress. In the real printing process, it reflects the shear stress/strain applied in the syringe to start the flow of the gelled solutions/suspensions.

Further measuring Pluronic solutions and glass inks in oscillatory mode allows to observe that, by increasing the percentage of Pluronic, the network becomes more elastic and flexible. The curves start to drop after a longer linear viscoelastic range, and this drop tends to occur at higher shear strain values with the increasing concentration of Pluronic solutions; however, for a fixed concentration of Pluronic, an increase of solid loading turns the network of the glass inks more brittle, as reflected by their gradually lower critical strains. Moreover, the values of elastic modulus (G′) grow with the increasing concentrations of both polymer solutions and glass inks, which agrees with the results obtained in the steady shear experiments. It is also worth mentioning that the cross-over points of G′ and G″ curves in Pluronic solutions change from 10% to 27%, while for the glass inks, they change from 27% to 17%, which show the converse trends, as shown in Figure 5. This means that the critical shear strain to break the gelled structure to flow is getting larger with the increasing concentrations of polymer, as the more micelles in the system, the more the network can deform before losing its structure. For the glass inks, the critical shear strain to break the ligation between the polymer and glass particles decreases with the increasing concentration of glass, because the glass particles are distributed homogeneously in the polymer solutions and the bonding is weaker between the particles and polymer chains than that among the polymer micelles, with a greater number of particles introduced, causing the critical shear strain to decrease. Different from the previous reports of the ink system consisting of dispersant–HPMC–PEI in [24,25], where a strain thinning behavior (type I) is shown, the ones of Pluronic solutions and glass inks exhibit a weak strain overshoot behavior (type III) [30], and their corresponding chain structures can be found in [31]. The different structures lie in the homogeneity of the as-mixed ink caused by the discrepancy of the particle size, that is to say, the glass powder still has larger particles than those in ceramic powder. Therefore, the oscillatory mode can offer more information than the rotational counterpart.

As mentioned above, the common dispersant–thickener–coagulant system is not applicable for 3D printing, owing to the insufficient stiffness of the as-printed layers for supporting the weight of the consecutive layers. The reason for this phenomenon is to account for the lack of ability to fast recover the gelled structure from the deformed state observed in the common ink system. In Figure 6, in order to analyze the change of the internal structure, the thixotropic behavior of both Pluronic solutions and glass inks is characterized in terms of the viscoelastic moduli by means of the three interval thixotropy test (3ITT) [31,32,33,34]. According to the results in Figure 5, where the shear strain spans from linear region to non-linear region, the input shear strain signal was chosen as 0.005% and 500%, respectively. For Pluronic solutions, in the first deformation step within the linear viscoelastic region (LVR), the G′ values are constant, which simulates that the structure is stable when the solutions/inks are not compressed. Then, the deformation enters into the non-linear region, as G′ immediately decreases, owing to the breakdown of the internal structure of the solutions/inks, simulating the printing process when the solutions/inks are under pressure. Subsequently, when the shear strain decreases back again to 0.005%, G′ gradually increases, as the internal structure recovers to the original gelled one when left standing, depicting the structural change in the as-printed filaments when attached to the substrate. The results from Pluronic solutions show good recovering ability, as G′ can reach 100% after sheared in the non-linear region. The glass inks inherit this excellent recovery behaviour from the Pluronic solutions, where G′ can rebuild up to 73.9%, 71.1%, and 97.2% after being sheared under 500% and the structure is left undeformed for 81 s, in terms of the increasing solids-loadings of the ink. When the solids-loading is 35 wt.% in the glass ink, it can even recover faster than the ones with the common dispersant–thickener–coagulant system reported previously [27].

Figure 7 presents DSC/TG curves of glass frit (a), printed sample (b) and pure Pluronic F-127 (c). In Figure 7a,b, one can see a drop of a heat flow in the temperature range below 50 °C. It is connected to higher inertia of the crucible with the sample in comparison with the reference. The reasons are ascribed to the high amount of powder specimen (around 50 mg) and high heating rate (10 °C/min). The DSC curve of pure polymer (Figure 7c) does not exhibit similar behavior because of a slower heating rate (2 °C/min). Measurement of pure glass frit revealed three inflection points above 500 °C. The first one at around 560 °C and second one at 690 °C may correspond to the glass transition temperature (T_g_), which means that the produced material is phase separated. The third inflection point at around 860 °C may represent the slow reaction occurring between the glass sample and crucible material. In the case of the printed sample, one can observe an endothermic peak at around 60 °C corresponding to the melting of Pluronic F-127, as it is also evident on the DSC curve of pure polymer. The strong exothermic peak at around 200 °C visible in both Figure 7b,c is an effect of polymer thermal decomposition. Contrary to Figure 7a,c, the printed sample exhibits an additional exothermic peak at around 310 °C. The most conceivable explanation is that the printed sample also contains residue of polytetrafluoroethylene, which is used as a construction material of attrition mill container. According to the DSC measurements, the debinding temperature was set at 500 °C for the best possible burnout of organics without allowing glass to flow over its T_g_.

The green printed samples are shown in Figure 8a, while samples obtained after the sintering process were still opaque (not shown in Figure 8a). In order to determine the source of scattering, SEM imaging and X-ray diffraction measurement were performed. Figure 8b presents an SEM micrograph of the fracture surface of the sintered glass sample. One can observe that the obtained cylinders are highly porous, which results in light scattering, while it has a homogeneous appearance. This fact implies that it is still necessary to improve the debinding and sintering parameters, or/and instead of processing in air atmosphere, apply vacuum for an effective deairing step.

Figure 9 shows the X-ray diffraction pattern of the sintered sample. This result allows to confirm that, after all steps of the manufacturing process, no crystalline phase was developed upon sintering. This result is consistent with the SEM images, where no additional phase was observed. However, further improvements in the proposed manufacturing process are required in the future in order to obtain transparent printed glass samples. Future works should focus on the following aspects to enhance the densification of the printed glass parts. In the ink mixing process, the composition of inks should be changed in order to increase volume fraction of the glass powder while preserving similar rheological properties so that the green samples should have lower initial porosity. Before printing, an efficient degassing step would be helpful for removing the air bubbles entrapped in the ink. Upon sintering, all the relevant process parameters should be suitably adjusted, and/or further tests exploiting sintering in vacuum should be performed.

## 4. Remarks and Conclusions

In this preliminary study, we successfully 3D printed water-sensitive and micron-sized borosilicate glass with high alkali and alkaline oxide contents by means of direct ink writing. Major difficulties reported in the literature, such as rheological instability of the ink or retaining the shape of the printed object during sintering process, were overcome. The main conclusions are as follows:(1)The designed manufacturing process is suitable for obtaining water-sensitive glass inks. The results of rheological measurements showed that inks based on Pluronic F-127 hydrogel are rheologically stable against glass dissolution.(2)The structural analysis after sintering of printed samples showed no crystalline phases introduced during manufacturing process. However, the obtained glass rings were opaque. SEM imaging revealed a significant amount of closed porosity, which may be responsible for light scattering. In order to obtain more transparent samples, ink composition should be enhanced, an additional degassing step should be added, or/and the sintering process should be improved.

## Figures and Tables

**Figure 1 materials-13-01636-f001:**
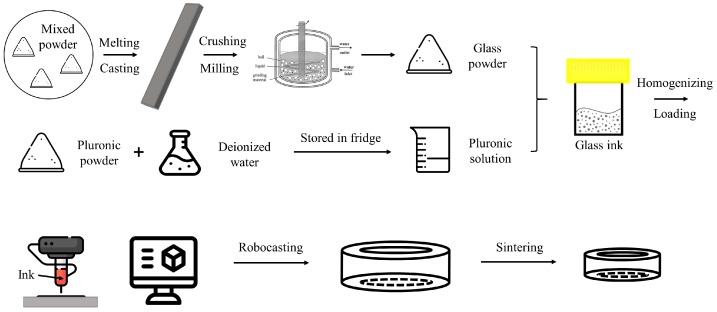
Schematics of glass ink preparation.

**Figure 2 materials-13-01636-f002:**
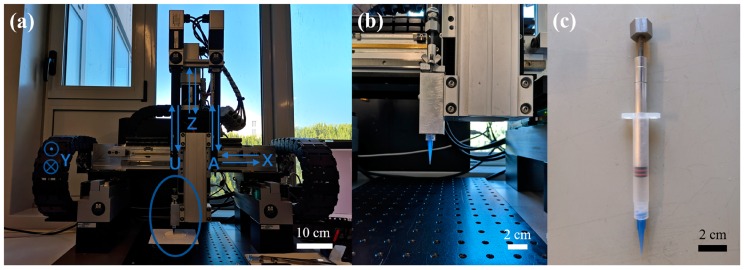
Facility of direct ink writing: (**a**) the Robocasting machine, (**b**) enlarged parts (left syringe chamber) as labelled in (**a**), and (**c**) a 3 mL syringe with a 410 μm nozzle.

**Figure 3 materials-13-01636-f003:**
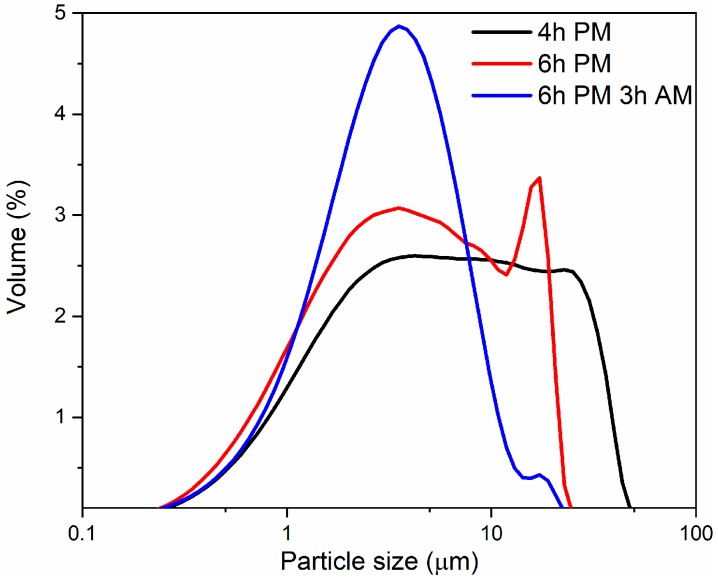
Particle size distributions of glass powders after different time of milling. Black line (4 h PM) corresponds powder milled in planetary mill for 4 h, green line (6 h PM) corresponds to 6 h of planetary milling (PM), blue line (6 h PM, 3 h AM) corresponds to powder obtained after 6 h of planetary milling followed by 3 h of attrition milling (AM).

**Figure 4 materials-13-01636-f004:**
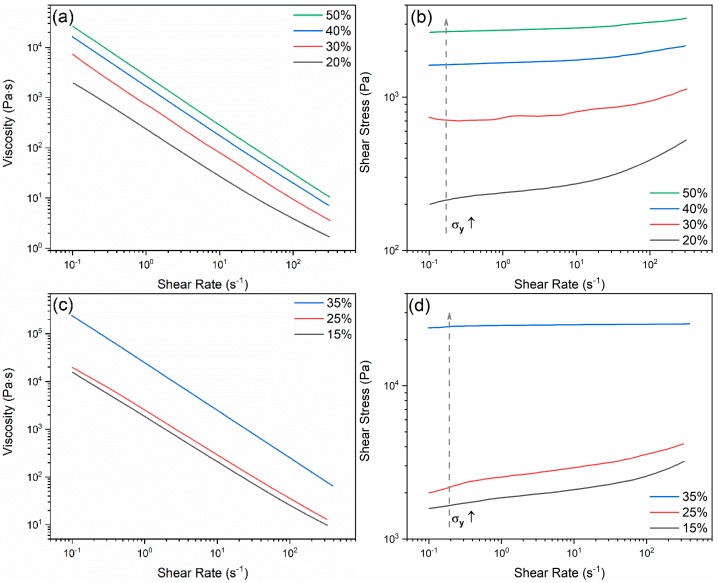
Flow curves and shear stress vs. shear rate curves of (**a**), (**b**) Pluronic solutions and (**c**), (**d**) glass inks of different solids loadings prepared in the 30 wt.% Pluronic stock solution.

**Figure 5 materials-13-01636-f005:**
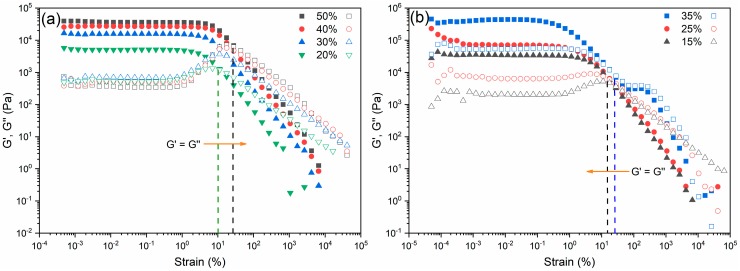
Linear viscoelastic region (LVR) of (**a**) Pluronic solutions and (**b**) glass inks measured under amplitude sweep mode. The elastic modulus (G′) and viscous modulus (G″), independently plotted in the same axis, are labelled as solid dots and empty symbols, respectively.

**Figure 6 materials-13-01636-f006:**
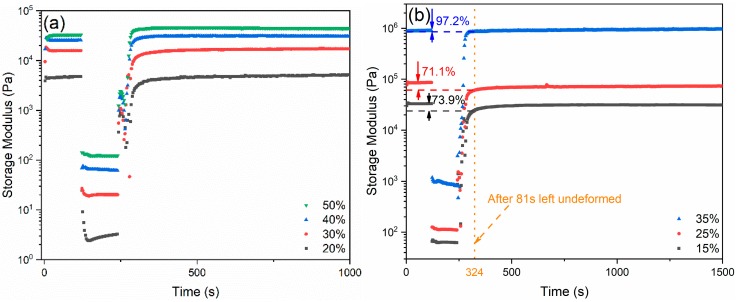
Three interval thixotropy test (3ITT) results describing changing structures of (**a**) Pluronic solutions and (**b**) glass inks.

**Figure 7 materials-13-01636-f007:**
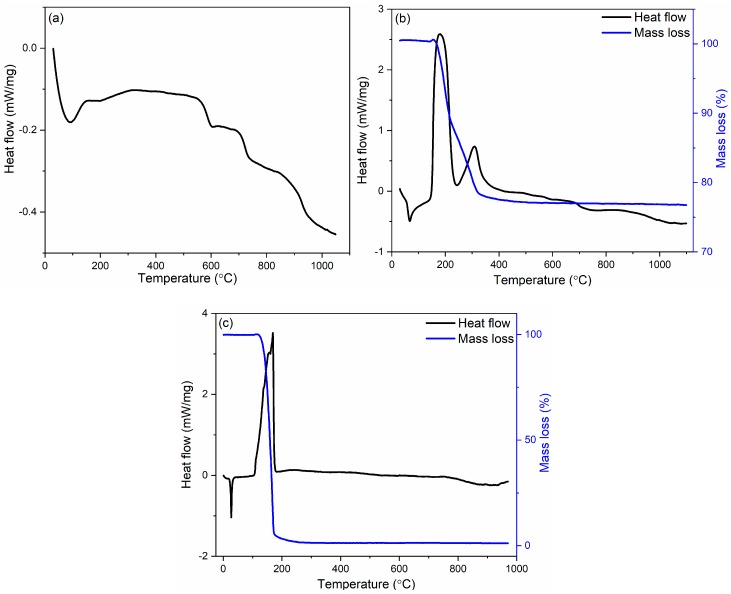
Differential scanning calorimetry coupled with thermogravimetry (DSC/TG) curves of borosilicate glass (**a**), printed sample (**b**), and pure Pluronic F-127 (**c**).

**Figure 8 materials-13-01636-f008:**
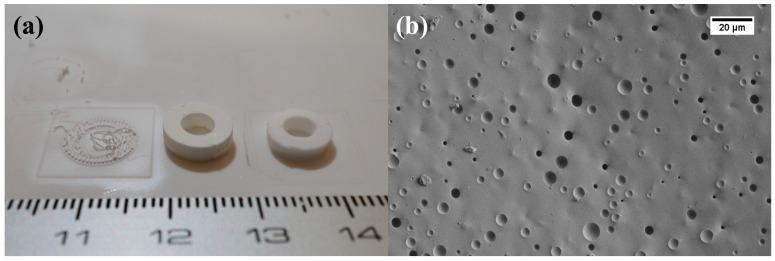
(**a**) Photo and (**b**) scanning electron microscope (SEM) micrograph of printed and sintered glass sample.

**Figure 9 materials-13-01636-f009:**
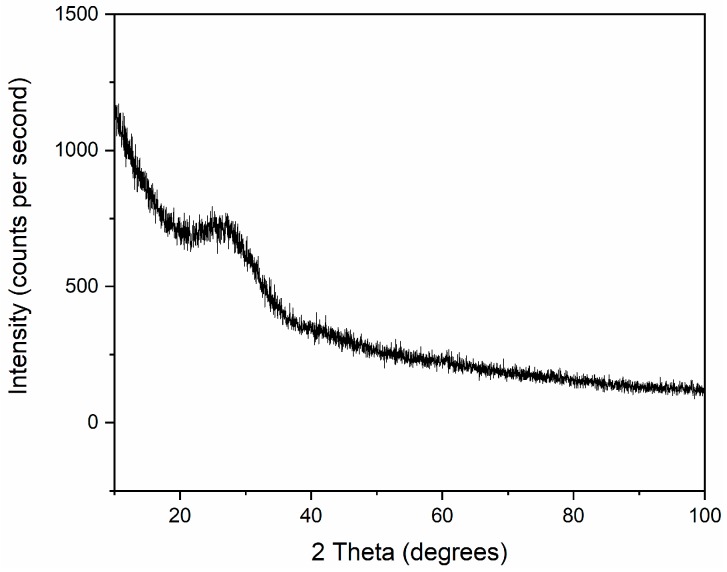
X-ray diffraction (XRD) pattern of printed and sintered glass sample.

**Table 1 materials-13-01636-t001:** Comparison of reported printing methods.

Method	Procedure	Printing Precursor(s)	Light-/Ink-Based	Nozzle	References
Molten filament deposition	One step	Molten glass	Ink-based	Yes	[18]
Laser aided filament deposition	One step	Glass filament	Both	Yes	[19]
Direct ink writing	More steps	Suspension of nanoparticles /Sol–gel derived feedstock	Ink-based	Yes	[12,14,15,16]
Stereolithography	More steps	Curable monomer with nanopowder	Light-based	No	[13]
Digital light processing	More steps	Alkoxide precursors, photoactive monomer and light-absorbing dye	Light-based	No	[9,17]

**Table 2 materials-13-01636-t002:** Chemical compositions of glass powder.

Compound	SiO_2_	Al_2_O_3_	B_2_O_3_	MgO	CaO	Na_2_O	K_2_O
**mole %**	62.5	2.0	11.5	7.0	4.0	3.5	9.5

**Table 3 materials-13-01636-t003:** Particle size distribution after different times of milling. PM, planetary milling; AM, attrition milling.

Sample	Particle Size Distribution (µm)				
	d_10_	d_25_	d_50_	d_75_	d_90_
**4 h PM**	1.2	2.4	6.0	15.2	26.7
**6 h PM**	1.0	1.9	4.2	9.4	16.1
**6 h PM, 3 h AM**	1.1	1.9	3.3	5.3	7.8

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
