# Peer review of "Direct Ink Writing Glass: A Preliminary Step for Optical Application"

_materials, 2020, doi:10.3390/ma13071636_

Round 1

Reviewer 1 Report

In this paper, the author present their research concerning the preparation and characterization of 3D printing material based on borosilicate glass in a Pluoronic hydrogel. They show that their designed material is suitable for its purpose, considering its rheological properties, and in spite of the propensity of the glass to undergo hydrolysis.

The authors provide a broad range of experiments to nicely support their conclusions, presented in a detailed way. Therefore, I recomment publication of the paper. However, some parts/contents of the paper should be improved.

  • The quality of the introduction is quite poor. There are very general statements which, in some cases are not completed, in other cases are unnecesary. For example:
    • Line 40: they mention that shaping methods can be categorized into seven types, but they do not mention the types. In any case, I do not think this is a relevant information.
    • Line 95 and afterwards: "while pH of the solution is changed". Changed to what, or in what sense? "The stability of hydrogel suspensions should increase". Increase with respect to that?
  • In results section, lines 217 and 219, the abreviation of planetary milling and attrition milling should be included in the text, since afterwards they appear in the figure. Also, the abreviation d50, and the other similiar dxx numbers in table 2 should be explained.
  • The explanation of the results in page 8 is very dense, and many explanations are only supported by text and not by figures. I think more figures (some of them in the form of insets) should be added. For example:
    • The yield stress mentioned in lines 264-266.
    • Some numbers and a figure should support the discussion on fragility and britleness as a function of concentration of Pluronic (lines 271 and afterwards).
    • The explanation of the crossover points in G' and G'' (lines 277 and around) should also be supported in a figure, specially because a conclusion is reached after this observation (line 278 and below).
    • There is an interesting discussion on recovery behaviour after printing in lines 312 and below. A graph (inset?) with this information may be useful in figure 6.
  • In figure 7, the heating rates (which are different among measurements) should be indicated.

Author Response

  • Q1: The quality of the introduction is quite poor. There are very general statements which, in some cases are not completed, in other cases are unnecesary. For example:

Line 40: they mention that shaping methods can be categorized into seven types, but they do not mention the types. In any case, I do not think this is a relevant information.

Line 95 and afterwards: "while pH of the solution is changed". Changed to what, or in what sense? "The stability of hydrogel suspensions should increase". Increase with respect to that?

A1: The authors agree that the information relative to the classification of new shaping methods into different types is not much relevant. However, it is provided for allowing the readers who have interest in further digging to consult the source of it - American Society for Testing and Materials (ASTM) 52900:2015 standard [1]. Because of that, we prefer to keep these sentences unchanged.

The stability of the Pluronic solutions against pH changes is well documented in the cited references. The idea is to underline that the stability of the Pluronic-based suspensions is not ionic strength sensitive, being therefore a wise choice for dispersing a somewhat water-soluble glass powder. Accordingly, the text was clarified.

  • Q2: In results section, lines 217 and 219, the abreviation of planetary milling and attrition milling should be included in the text, since afterwards they appear in the figure. Also, the abreviation d50, and the other similiar dxx numbers in table 2 should be explained.

A2: The abbreviations for planetary milling (PM) and attrition milling (AM) were already defined in the section “2. Materials and Methods”, where they were firstly mentioned. For d50 and other related numbers, the explanation of the abbreviations is added in lines 221-223.

  • Q3: The explanation of the results in page 8 is very dense, and many explanations are only supported by text and not by figures. I think more figures (some of them in the form of insets) should be added. For example:
    • The yield stress mentioned in lines 264-266.
    • Some numbers and a figure should support the discussion on fragility and britleness as a function of concentration of Pluronic (lines 271 and afterwards).
    • The explanation of the crossover points in G' and G'' (lines 277 and around) should also be supported in a figure, specially because a conclusion is reached after this observation (line 278 and below).
    • There is an interesting discussion on recovery behaviour after printing in lines 312 and below. A graph (inset?) with this information may be useful in figure 6.
  • In figure 7, the heating rates (which are different among measurements) should be indicated.

A3: a. For the yield stress, one can roughly read the values from Figure 4(b) and (d). In our analysis of the results, the exact value of the yield stress is not relevant, but the fact that it increases with the concentration. We modified figures 4(b) and 4(d) introducing an arrow that indicates how the yield stress value increases with the concentration.

  1. Actually, the values of the shear strain have been referred to explain and support the discussion about the brittleness of the Pluronic in a later context. The new version of the manuscript contains a more detailed description for the sake of clarity.
  2. A new Figure 5 has been introduced in the revised version of the manuscript in order to explain clearly how the cross-over point changes with the increase of the concentrations of Pluronic or glass powder.
  3. The recovery was already clearly shown in Figure 6. Even though, Fig. 6(b) was improved to turn things clearer.
  4. The heating rates, which were described in the section “2. Materials and Methods”, are now also mentioned in the text close to the figures: 10°C/min (Fig. 7(a), Fig. 7(b)), and 2°C/min (Fig. 7(c)).

Reviewer 2 Report

In the paper "Direct ink writing glass: a preliminary step for optical application" by B. Nan et al. the authors describe a protocol to 3D printing a borosilicate glass ink starting from grinded micron-sized particles and a matrix of Plurionic F-127 hydrogel. 

The procedure followed in the paper is very well described and detailed. The characterization is completed and well-executed.

My minor comments are: 

  • Can the authors comment on the geometrical limitation of this technique with the proposed material? Is there a practical limit or prohibited geometry in the technique, given by the heat treatment after printing?
  • Please include the experimental details of the SEM characterization including experimental conditions and sample preparation. 
  • Is it possible, changing the protocol, to obtain an open porosity to envision applications in gas storage and biomaterials?   

Author Response

  • Q1: Can the authors comment on the geometrical limitation of this technique with the proposed material? Is there a practical limit or prohibited geometry in the technique, given by the heat treatment after printing?

A1: Good question. Yes, for sure there is limitation in this technique for this material. For example, the printing results can be incomplete or sometimes disastrous for a geometry with a strange center of gravity or for some parts of a geometry suspending in the air, even if there is some supporting material. The reason can be ascribed to the increasing mass of the already printed material that might overcome the limited strength of the support material. However, the same geometry might be printable without any problem using another material system and the shape might be maintained as printed after heat treatment. So far, we didn’t see the limitation of the geometry when a proper ink system was utilized, as we’ve tried to print many kinds of complex shapes.

  • Q2: Please include the experimental details of the SEM characterization including experimental conditions and sample preparation. 

A2: Thanks for remark. All the experimental details of the SEM characterization were included in the revised manuscript in the section “2. Materials and Methods”.

  • Q3: Is it possible, changing the protocol, to obtain an open porosity to envision applications in gas storage and biomaterials?   

A3: Yes, it is possible. Actually, our research group has good experience in printing porous bioglass scaffolds for bone regeneration and tissue engineering.

And there should not be any problem in extending the same approaches to gas storage applications.

Reviewer 3 Report

Bo Nan et al. described the preliminary study of 3D printing with the use of micron-sized borosilicate glass. Some difficulties, which has been reported in the literature, were overcome. Among them are rheological instability of the glass mixture as well as retaining the object shape after the sintering process. Appropriate graphs and tables supported all experiments. Provided explanations and findings are accurate and enable their understanding. However, at the current stage, the manuscript requires some amendments:

1) The first appearance of abbreviations: DSC, SEM and XRD in the abstract have not been extended, so they are unreadable at the beginning of the manuscript.

2) The authors forgot to explain what means "HPMC-PEI".

3) G'/G" in Figure 5 is misleading - because it implies a quotient, and that is not what the authors suggest.

4) No legends are described for figures 5 and 6.

Moreover, the reviewer does not agree with the statement: "This implies that the proposed manufacturing process is suitable for the preparation of printed transparent glass samples after the implementation of any possible improvements... ". It is unjustified - the authors did not show a transparent glass printout.

Author Response

  • Q1) The first appearance of abbreviations: DSC, SEM and XRD in the abstract have not been extended, so they are unreadable at the beginning of the manuscript.

A1: The abbreviations were explained in the abstract.

  • Q2) The authors forgot to explain what means "HPMC-PEI".

A2: The necessary corrections were made in the revised manuscript.

  • Q3) G'/G" in Figure 5 is misleading - because it implies a quotient, and that is not what the authors suggest.

A3: Here, the slash “/” did not stand for a mathematical division. It was used just to separate and indicate that there were two parameters (G’ and G’’) plotted on the same y-axis of Figure 5. Nevertheless, a new Figure 5 has been included in the revised version of the manuscript where “G’, G’’ is used instead of “G’/G’’. To make it clearer, this is also stated in the caption of Figure 5.

  • Q4) No legends are described for figures 5 and 6.

A4: We do not know what may have happened with your version of the manuscript, but the legends of these figures were already displayed.

  • Q extra: Moreover, the reviewer does not agree with the statement: "This implies that the proposed manufacturing process is suitable for the preparation of printed transparent glass samples after the implementation of any possible improvements... ". It is unjustified - the authors did not show a transparent glass printout.

Extra A: Yes, there were some difficulties in obtaining the transparent result. However, as this is a conceptual draft, we are currently working in using a sol-gel technique for the preparation of the raw powder, to get a better output.

Round 2

Reviewer 1 Report

The corrections are sufficient to me. I recommend publication as it is now.